# Complex II Biology in Aging, Health, and Disease

**DOI:** 10.3390/antiox12071477

**Published:** 2023-07-24

**Authors:** Eric Goetzman, Zhenwei Gong, Bob Zhang, Radhika Muzumdar

**Affiliations:** 1Division of Genetic and Genomic Medicine, Department of Pediatrics, University of Pittsburgh, Pittsburgh, PA 15260, USA; 2Division of Endocrinology, Department of Pediatrics, University of Pittsburgh, Pittsburgh, PA 15260, USA

**Keywords:** aging, Complex II, succinate dehydrogenase, reactive oxygen species

## Abstract

Aging is associated with a decline in mitochondrial function which may contribute to age-related diseases such as neurodegeneration, cancer, and cardiovascular diseases. Recently, mitochondrial Complex II has emerged as an important player in the aging process. Mitochondrial Complex II converts succinate to fumarate and plays an essential role in both the tricarboxylic acid (TCA) cycle and the electron transport chain (ETC). The dysfunction of Complex II not only limits mitochondrial energy production; it may also promote oxidative stress, contributing, over time, to cellular damage, aging, and disease. Intriguingly, succinate, the substrate for Complex II which accumulates during mitochondrial dysfunction, has been shown to have widespread effects as a signaling molecule. Here, we review recent advances related to understanding the function of Complex II, succinate signaling, and their combined roles in aging and aging-related diseases.

## 1. Introduction

Mitochondria play a critical role in cellular respiration and energy metabolism. The mitochondrial respiratory chain is a source of ATP. It is composed of a series of protein complexes in the inner mitochondrial membrane. Mitochondrial Complex II, also known as succinate dehydrogenase (SDH), is one of the four complexes of the respiratory chain. It plays a critical role in the tricarboxylic acid (TCA) cycle, a key metabolic pathway involved in the oxidation of carbohydrates, lipids, and amino acids. Complex II catalyzes the oxidation of succinate to fumarate. The electrons produced during this reaction are transferred to the electron transport chain (ETC), where they are used to generate ATP. In this review, we will discuss the structure and functions of Complex II and its role in aging and age-associated diseases.

## 2. Complex II: Structure and Function

### 2.1. Overall Structure and Assembly

The crystal structure of mammalian Complex II was first solved in 2005 for the porcine complex [1]. In 2023, the structure of human Complex II was also solved [2]. Complex II is small relative to other electron transport chain complexes, with a total size of 124 kDa. In shape, the overall structure resembles a flower, with a large hydrophilic head protruding into the mitochondrial matrix and a hydrophobic stem anchored in the inner mitochondrial membrane (Figure 1). Complex II consists of four subunits named SDHA, SDHB, SDHC, and SDHD. The hydrophilic head is composed of the SDHA subunit (68 kDa) and SDHB subunit (29 kDa), and the hydrophobic stem is made of SDHC (15 kDa) and SDHD (11 kDa). A small portion of the hydrophobic stem protrudes into the intermembrane space. Unlike other respiratory chain complexes, electron transfer through SDH occurs without proton pumping. The catalytic site for Complex II resides on SDHA, a flavoprotein that reversibly converts succinate to fumarate. SDHA extracts electrons and channels them from its flavin adenine dinucleotide (FAD) cofactor to the neighboring subunit SDHB, which contains three iron sulfur clusters. SDHC and SDHD contain two ubiquinone binding sites, one on the matrix side of the inner mitochondrial membrane, and one on the intermembrane space side of the membrane. It is thought that only the site on the matrix side plays a functional role [3]. SDHC and SDHD also coordinate a heme cofactor. It remains unclear whether this heme plays a role in the electron transfer to ubiquinone, but it may help stabilize the complex [1,3,4].

Unlike other respiratory chain complexes, Complex II is completely encoded in the nucleus, with no contribution from mtDNA. Over the past 15 years, details have begun to emerge regarding the assembly of Complex II and the participation of assembly factors [3,5,6]. There are four assembly factors (AF) involved in the formation of Complex II, known as SDHAF1 to SDHAF4. SDHAF1 and SDHAF3 assist with the maturation of SDHB and inserting the iron sulfur clusters while shielding them from oxidative damage. SDHAF2 is a chaperone that promotes flavination of the SDHA subunit, working in concert with SDHAF4, which protects the FAD from the solvent until SDHA and SDHB dimerize. The role these assembly factors play in health and disease remains an ongoing area of investigation. Intriguingly, an alternative, non-membrane-bound, soluble form of Complex II consisting of SDHA in a complex with its two assembly factors, SDHAF2 and SDHAF4, was recently observed in the mitochondria under bioenergetic stress [7].

### 2.2. Inhibitors and Production of Reactive Oxygen Species (ROS)

Much of what is known about Complex II’s role in metabolism comes from studies leveraging inhibitors. Inhibitors of Complex II can be divided into two classes: those that bind to the active site on SDHA, and those that disrupt the ubiquinone binding pocket [4,8,9,10]. Malonate is a three-carbon dicarboxylic acid that competitively inhibits Complex II at the succinate binding site. Oxaloacetate, a Kreb cycle intermediate downstream of Complex II, also exhibits competitive inhibition at the succinate binding site. One widely used inhibitor that irreversibly binds to the SDHA active site is 3-nitropropionic acid (3-NP). However, fungicides such as thenoyltrifluoroacetone (TTFA), boscalid, bixafen, carboxin, fluopyram, promysalin, diazoxide, atpenin A5, and siccanin target the ubiquinone-binding site. These fungicides are commonly applied in agriculture, exhibiting mild environmental side effects.

The fungicide inhibitors cause electron slippage at the ubiquinone binding site and subsequent superoxide production. While this is accepted as fact now, for decades, Complex II was disregarded as a potential source of reactive oxygen species (ROS). The discovery in the early-to-mid 2000s that Complex II mutations cause significant ROS in the context of various cancers led to a subsequent re-examination of the role of Complex II in ROS biology. As recently reviewed in detail by Vanova et al. [11], over the past decade, it has become clear that Complex II can indeed contribute to cellular ROS production and does so under both normal and diseased conditions (Figure 2). In the normal, healthy state, mitochondrial concentrations of succinate, the substrate for Complex II, remain at ~0.5 mM or below. In this context, the flavoprotein SDHA is frequently in a redox state where it has accepted electrons from succinate and released the product fumarate, but the electrons are still held by the flavin rings. Many flavoproteins, when left unshielded in the reduced state, will leak electrons to oxygen [12,13,14]. It turns out that SDHA is also subject to leaking electrons from the FAD cofactor when the substrate-binding pocket is empty and oxygenated solvent has access to the active site [15]. Gene mutations that impair Complex II assembly or disrupt Complex II subunit–subunit interactions can exacerbate this leak. Paradoxically, this form of ROS production can also become suppressed under certain disease states. One notable example is ischemia–reperfusion injury. During ischemia, succinate levels can rise dramatically inside the mitochondria, fully saturating the SDHA active site and protecting the reduced FAD cofactor from oxygen. While this disease state efficiently prevents direct ROS formation by Complex II, it unfortunately leads to a much more potent and damaging source of ROS, known as reverse electron transfer (RET). Upon reperfusion, the sudden massive wave of electrons flooding through Complex II from succinate oxidation leads to a rapid and complete reduction of the ubiquinone pool, forcing electrons backward from ubiquinone to Complex I, from which they are leaked to oxygen in large quantities.

## 3. Complex II and Disease

In short, it is now understood that Complex II is quite complex indeed. In normal bioenergetic states, it functions primarily as a succinate dehydrogenase enzyme and secondarily as a source of healthy ROS. In some circumstances, the complex may run in reverse, reducing fumarate to succinate. When any of these normal functions go awry, disease and unhealthy aging may ensue. This is evidenced by the increased risk for certain types of tumors with mutations of the Complex II-related genes. Loss of function due to genetic or acquired causes have been shown to cause poor oxidative phosphorylation, generation of excessive ROS, and tumor predisposition. Levels of Complex II have been shown to be decreased in genetic disorders and in normal aging, and have been associated with many age-related disorders. In the sections below, we will discuss these in detail.

### 3.1. Inherited Disorders of Complex II Dysfunction

Deficiencies in Complex II are rare and have been associated with a few inherited disorders and tumor disposition syndromes. The clinical phenotype of Complex II deficiency varies based on the subunit that houses the mutation [16]. SDHA is the major catalytic subunit of SDH and is the most stable of SDH proteins. Mutations in SDHA are rare and vary in clinical phenotype depending on the extent of loss of Complex II activity in specific tissues. Homozygous or compound heterozygous mutations in the SDHA gene can result in Leigh’s syndrome, a progressive neurodegenerative disorder which can result from mutations in Complex I–V of the respiratory chain, Coenzyme Q, or pyruvate dehydrogenase (PDH) complexes [17]. Key clinical features of Leigh’s syndrome include psychomotor delay, regression, hypotonia, ataxia, and respiratory insufficiency. A key biochemical feature is lactic acidemia. Specific magnetic resonance imaging (MRI) observations include lesions in cerebral white matter and basal ganglia. The frequency and clinical phenotype of individual mutations are highly varied, with certain mutations causing progressive disease while some show stability and improvement. Certain mutations are only associated with cardiomyopathy while others can have both neurological and cardiac phenotypes. Dilated cardiomyopathy could also be an isolated feature without neurological involvement [18]. Though the differences in disease symptoms could be attributed to tissue-specific expression and variable penetrance, the exact underlying reasons are still unknown. A heterozygous mutation amounting to a 50% reduction in Complex II activity may develop a late onset neurodegenerative disease. Infantile leukodystrophy phenotype along with late-onset optic atrophy, ataxia, and myopathy have been reported. Of note, parents who are carriers of the heterozygous mutation could remain clinically asymptomatic [16].

SDHA–D and AF2 have been reported to have tumor suppressor functions. Mutations in these can lead to tumor predisposition syndrome. Mutations in SDH also increase oncogenic potential. Succinate accumulation, inhibition of alpha ketoglutarate (α-KG)-dependent dioxygenases, generation of ROS, inhibition of histone demethylases, stabilization/activation of HIF1- alpha, and inhibition of EglN3-mediated apoptosis have all been proposed as mechanisms to account for the increased tumorigenesis with SDH [19]. Indeed, ~15% of paragangliomas/pheochromocytomas are attributable to mutations in Complex II (SDHA, B, C, D, and AF2). SDHA mutations have also been reported in gastrointestinal stromal (GIST) tumors [16]. Germ line mutations in SDHB, C, and D genes have been reported in Carney–Stratakis syndrome [20]. SDHB is the most commonly mutated of all the SDH-related genes, with >180 mutations that have been associated with extra adrenal, head or neck paraganglioma, adrenal pheochromocytoma, Carney–Stratakis syndrome, renal cell carcinoma, GIST, and infantile leukodystrophy. The penetrance is low, with only 25–40% of carriers developing a tumor in their lifetime. The reasons for the low penetrance are still unclear. However, ~20% of carriers will develop metastatic disease [21,22].

SDHD mutations are the second most common after SDHB. They show a unique and interesting inheritance pattern and resemble imprinted disorders. Carriers of SDHD mutations will only develop tumors when the mutation is inherited from the father [23]. Tumors are rare if the mutation is inherited from the mother [24]. This is also observed in SDHAF2 mutations, and, interestingly, SDHD and SDHAF2 share a chromosomal location (chromosome 11) [23]. SDHD mutations have a high penetrance, with 87–100% of carriers showing tumor development, though many may not have overt clinical symptoms. Mutations in SDHD have been associated with benign head and neck paragangliomas, adrenal pheochromocytomas, Carney-Stratakis syndrome and GIST tumors. SDHC mutations have low penetrance and are a rare cause of paraganglioma.

### 3.2. Complex II in Aging-Associated Neurodegenerative Diseases

It is well-known that aging is associated with a decline in mitochondrial function, which may contribute to age-related diseases, including neurodegenerative diseases, cancer, and cardiovascular diseases [25]. Complex II has emerged as an important player in the aging process with its recognized role in metabolism and oxidative stress. A growing body of evidence suggests that Complex II dysfunction contributes to age-related diseases [26]. Complex II activity has been shown to decline with age in multiple tissues, including the brain, liver, heart, and skin [27,28,29,30]. The age-associated decline in complex II activity may contribute to the accumulation of mitochondrial DNA (mtDNA) mutations, oxidative damage, and changes in the expression of genes involved in mitochondrial biogenesis [31]. The decline in Complex II activity is further associated with a decrease in ATP production and an increase in the production of ROS, which can lead to oxidative stress and cellular dysfunction through damage to proteins, lipids, and DNA [8,11].

In particular, reduced activity of the Complex II enzyme has been observed in many neurodegenerative diseases. For example, Complex II activity has been found to be significantly reduced in the brain of patients with Alzheimer’s disease (AD) [32]. A similar reduction of Complex II activity was also observed in the brain of patients with Parkinson’s disease (PD), and mutations in the genes encoding Complex II subunits have been linked to PD [33]. Interestingly, inhibition of succinate dehydrogenase using 3-nitropropionic acid induced a clinical and pathological phenotype of Huntington disease (HD) in non-human primates, suggesting that Complex II may also play a role in HD [34]. Furthermore, overexpression of Complex II subunits restored Complex II levels and mitochondria function and blocked striatal cell death [35].

### 3.3. Complex II Dysfunction in Metabolic Syndrome

In addition to neurodegenerative disorders, several studies have shown that the activity of Complex II is reduced in other age-related metabolic diseases, such as obesity and type 2 diabetes [36,37]. It has been shown that visceral adipose tissue exhibited decreased Complex II activity compared to the subcutaneous adipose tissue, and weight loss improved Complex II activity [36]. A small-molecule compound that induces complex II subunit expression improved obesity-associated disorders [38]. The molecular mechanisms underlying the relationship between mitochondrial Complex II and age-related metabolic diseases are complex and not fully understood. However, several potential mechanisms have been proposed based on preclinical and clinical studies. Studies have suggested that the metabolic sensor, AMP-activated protein kinase (AMPK), is involved in the regulation of Complex II activity [39,40]. AMPK activation has been shown to increase cellular metabolism, leading to improved cell survival, while blockage of Complex II activity using an inhibitor completely abolished this effect [39]. Acute exercise and the AMPK activator, AICAR, induces Complex II activity in wild-type but not AMPK knockout mice [41]. Interestingly, the activation of AMPK decreases with age [42]. 

Obesity and metabolic syndrome are known to be associated with a state of chronic inflammation [43]. Circulating proinflammatory cytokines such as tumor necrosis factor-alpha (TNF-α) and interleukin-6 (IL-6) are increased in obese subjects, likely due to increased secretion by adipose tissue [43,44]. These cytokines have been shown to suppress mitochondrial respiration in non-adipose tissues, such as the heart, via multiple respiratory chain complexes, including Complex II [45,46]. This may cause a vicious cycle, as inhibition of Complex II leads to accumulation and release of succinate, which, in turn, further stimulates a proinflammatory state in both macrophages and T-cells [47,48]. 

### 3.4. Complex II and Succinate in Cardiac Disease

Among the age-associated disorders, cardiovascular disorders are the leading cause of mortality and morbidity. Both ischemia and reperfusion injury are associated with significant ROS-related damage to the mitochondrial respiratory chain machinery and other metabolic components, causing further ROS production and a vicious cycle of bioenergetic decline that leads to cell death. The role of Complex II in myocardial ischemia–reperfusion (MI–R) has been widely investigated. Complex II, through mitochondrial metabolism, links energy production to cell survival during ischemia. A stable Complex II is necessary to couple mitochondrial respiratory reserve capacity to increased TCA flux and ETC activity. Many studies have shown that ischemia leads to suppression of all complexes of the electron transport chain, with NADH dehydrogenase being most susceptible [49]. The impairment of Complex II results in changes in cell survival, ROS, and cellular bioenergetics, factors that affect tissue viability and outcomes after an ischemic event.

Mitochondrial and metabolic changes happen very quickly during MI–R. Ischemia and oxygen deprivation can lead to swelling of mitochondria and fragmentation of the inner membrane within minutes [50]. As early as two minutes into ischemia, glutamate dependent respiration is compromised along with malate supported respiration. The decrease can be noted in Complexes I, III, and IV. By 30 min, malate dependent respiration is significantly suppressed compared to succinate supported oxygen consumption [51]. One of the well characterized changes is the accumulation of succinate in the myocardium during ischemia. The ischemic accumulation of succinate has been reported across diverse species and tissues [52,53,54] and can be dramatic (more than 30-fold increase in the levels) [52,55,56,57]. An increase in plasma succinate following ischemia/exercise has also been reported, and this efflux of succinate from the ischemic myocardium is mediated through monocarboxylate transporter-1 (MCT1) [58]. 

There are many pathways that can contribute to the succinate pool under ischemic conditions. Zhang et al. [59] showed that the source of this succinate in the ischemic myocardium is the TCA cycle, rather than Complex II inhibition. Increased lactate accumulation through anaerobic glycolysis serves as a carbon source when lactate enters the cardiomyocyte through MCT1, is converted to pyruvate, and enters the TCA cycle [60]. There is also increased conversion of pyruvate to alanine, coupled with increased succinate from αKG via aminotransferases. The decrease in αKG during ischemia is replenished from glutamate, which is derived from glutamine (anaplerosis). All these metabolic shifts, along with pyruvate entering TCA cycle through oxaloacetate, result in increased succinate production. In addition to the canonical Krebs cycle, ischemia-induced reduction of Complex I drives succinate production from fumarate through the enzymatic reversal of Complex II. This facilitates glycolysis and proton pumping of complex I. This reversal of Complex II, which can be a source of reactive oxygen species, has been proposed but has not been explicitly demonstrated. In fact, some argue that an in situ reversal is not favored by the properties of Complex II. 

Irrespective of the source, an accumulation of succinate could be a double-edged sword in MI–R; while it could be beneficial and serve as an energy source during ischemia, it could also be a source of ROS and induce cardiac damage during reperfusion. The increase in plasma succinate, facilitated by the acidification of the myocardium that occurs following ischemia [58], is considered a mitokine as it signals hypoxia and distress to other tissues. Succinate binds to its own receptor on the vasculature and plays a role in maintaining vasomotor tone [61,62]. These observations supported studies that targeted succinate metabolism in MI–R injury. Indeed, the SDH inhibitor malonate, given in its prodrug form, dimethyl malonate, before or during ischemia and at reperfusion, has been shown to be cardioprotective in a pig model of MI–R [63]. Similarly, SDH inhibition has been shown to be effective in ischemic stroke [64] and kidney ischemia–reperfusion [65]. Dimethyl malonate decreased brain damage following resuscitation of a cardiac arrest in rats [66]. 

Dimethyl malonate offers several advantages, including its ability to enter mitochondria, the requisite site of action. Dimethyl malonate also has very limited toxicity. However, a few challenges remain to be solved before dimethyl malonate can be used therapeutically. First, since it is a competitive inhibitor, remarkably high concentrations are needed. Second, the drug has to be targeted only to the ischemic tissues to avoid off-target effects [67]. The latter was recently addressed by a study that showed that malonate’s entry into the cardiomyocyte occurs through MCT1 transporters and is facilitated by lower pH during reperfusion, thus allowing selective targeting of at-risk tissues [68]. Remarkably, the metabolic reprogramming induced by dimethyl malonate has been shown to stimulate adult cardiomyocytes to re-enter the cell cycle and undergo mitosis in a mouse model of myocardial infarction [69]. This raises the exciting possibility that malonate may be used to promote heart regeneration after injury [69]. It may also remain effective in the presence of comorbidities such as obesity and diabetes, as evidenced by one preclinical study showing that malonate offered cardioprotection in Zucker diabetic rats, albeit at higher doses compared to non-diabetic rats [70].

## 4. Succinate as an Endocrine/Paracrine Factor

When Complex II is inhibited, such as by dimethyl malonate as detailed above, the resulting succinate accumulation is now understood to have much broader physiological effects than previously thought due to succinate export into circulation. Over the last 10 years, evidence has emerged showing that extracellular succinate is a powerful signaling molecule. The mean plasma level of succinate is normally about 5 µM [71], but this level can change during events such as ischemia or exercise, coinciding with oxidative stress or low energy level [62,72,73]. Succinate interacts with the renin–angiotensin–aldosterone pathway in modulating blood pressure level [73]. Succinate also plays a role in the lipolysis of adipose tissues [74,75]. The physiological effects of succinate depend on its binding with the cell-surface succinate receptor SUCNR1, formerly known as GPR91. SUCNR1 is a member of the δ Rhodopsin-like G protein-coupled receptors (GPCR) family [76]. The general structure of a GPCR consists of seven transmembrane regions or polypeptide chains with an extracellular N-terminus and intracellular C-terminus. He et al., through mutagenesis experiments, determined that the binding of succinate onto the Arg 99, His 103, Arg 252, or Arg 281 residues within the ligand-binding pocket results in the activation of SUCNR1 [71]. This causes a conformational change at the cytoplasmic domain that leads to heterotrimeric G protein activation. SUCNR1 can be found in several tissues and cell types, including adipose tissue, liver, kidney, pancreas, spleen, skin, brain, lymph nodes, gall bladder, thymus, and intestine [72,75]. It is also expressed in bone marrow, heart, ovary, and prostate tissue in very low amounts. SUCNR1 is expressed on hepatic stellate cells in the liver [77]. It can also be found in non-myofibrillar cells in muscle tissue, particularly satellite cells [78]. Within the central nervous system, it can be detected in cortical neurons, retinal ganglion cells (RGC5), and astrocytes [56]. Finally, the receptor has been observed on immune cells, including T-cells, B-cells, macrophages, and immature dendritic cells [79]. 

The effects of the succinate signaling pathway remain only partially characterized, but it is a promising future target for diseases related to aging. A decade ago, Favret et al. showed that SUCNR1 expression in the retinal pigment epithelium, declines dramatically in mice during aging, leading to outer retinal lesions [80]. In muscle, exercise causes the release of succinate, which then activates neighboring satellite cells in a paracrine fashion, leading to the remodeling of muscle tissue into a more mitochondria-rich, oxidative phenotype, with increased innervation [78]. This raises the possibility that stimulating this mechanism exogenously could help maintain muscle function and metabolism as we age, and to counter sarcopenia. However, whole-body stimulation of SUCNR1 may not be beneficial and would need to be conducted in a tissue-specific manner. For example, in adipose tissue, succinate activation of SUCRN1 suppresses lipolysis [75]. Circulating succinate increases during the fasted-to-fed transition and acts to help shut down lipolysis. In this case, inhibiting SUCRN1 could be desirable, promoting lipolysis and limiting adipose tissue expansion. Likewise, stimulation of SUCNR1 on hepatic stellate cells may promote fibrosis and non-alcoholic steatohepatitis (NASH), again suggesting a benefit to blocking succinate signaling. The signaling aspects of succinate biology will continue to be an active area of investigation in the next decade.

## 5. Conclusions

Complex II holds a unique place in bioenergetics, operating at the crossroads of the TCA cycle and the electron transport chain. We now understand much more about how Complex II contributes to both health and disease. Complex II declines during aging; thus, there is great interest in boosting its function to prevent aging-related disease. However, on the other hand, there are also many instances where too much Complex II activity is pathologic, and in those cases, inhibitors of Complex II are beneficial. Whereas key studies over the past 20 years greatly deepened our understanding of the context-specific biology of Complex II, the next 20 years are likely to usher in a new era of pharmacologic manipulation of Complex II to combat disease and extend the health span.

## Figures and Tables

**Figure 1 antioxidants-12-01477-f001:**
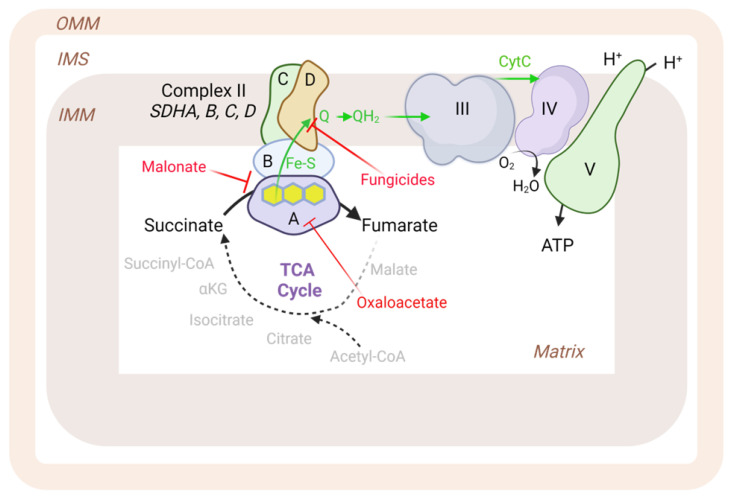
Structure and functional role of mitochondrial Complex II. Complex II is composed of four nuclear-encoded subunits SDHA, SDHB, SDHC, and SDHD (labeled “A”, “B”, “C”, and “D” in the figure).Under normal physiological conditions, SDHA serves to oxidize succinate to fumarate, transporting electrons first to the FAD cofactor on SDHA (depicted in yellow rings), then to a series of three iron-sulfur clusters (Fe-S) embedded in SDHB, and ending with transfer to ubiquinone (Q) at the interface of SDHC/D and the inner mitochondrial membrane (IMM). Malonate and oxaloacetate are naturally occurring inhibitors that compete with succinate for binding to the active site. A series of fungicides are known Complex II inhibitors via the ubiquinone binding site. Other abbreviations: OMM, outer mitochondrial membrane; IMS, inter-membrane space; III, respiratory chain Complex III; IV, respiratory chain Complex IV; V, respiratory chain Complex V (ATP synthase); CytC, cytochrome C; TCA, tricarboxylic acid cycle. Figure created with BioRender.com (accessed on 9 June 2023).

**Figure 2 antioxidants-12-01477-f002:**
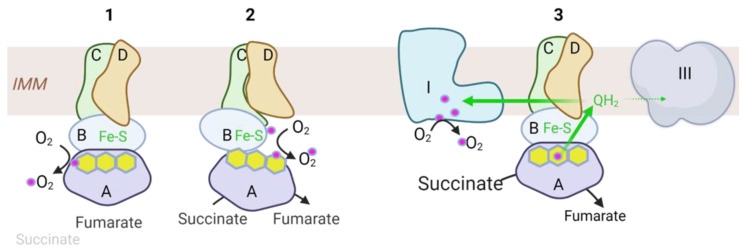
Complex II produces ROS under three contexts. (1) When succinate is at normal physiological concentrations, the binding site is often left unoccupied while the FAD cofactor is in the reduced state (electrons indicated as pink circles). This allows oxygenated solvent access to steal electrons from the FAD cofactor. (2) Inborn errors of Complex II genes, Complex II assembly factors, or acquired mutations in cancer can disrupt the functional integrity of the complex, promoting electron leak from either the FAD or from the iron-sulfur (Fe-S) clusters. (3) Reverse electron transport can be a robust source of ROS under conditions of high succinate concentrations. Here, succinate saturates the SDHA biding site, preventing oxygen from stealing electrons, but the ubiquinone system cannot keep up and becomes highly reduced (QH2), leading to electrons flowing upstream to Complex I, where they react with oxygen to form ROS. Other abbreviations: IMM, inner mitochondrial membrane; III, respiratory chain Complex III; A,B,C,D, succinate dehydrogenase subunits SDHA, SDHB, SDHC, and SDHD. Figure created with BioRender.com (accessed on 9 June 2023).

## Data Availability

Not applicable.

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
