# Peer review of "Complex II Biology in Aging, Health, and Disease"

_antioxidants, 2023, doi:10.3390/antiox12071477_

Round 1
Reviewer 1 Report
The authors provided a comprehensive and scientific-sound review to introduce functions of complex II in ETC and the roles of complex II in diseases. It only need to have minor revised in order to publish.
1. Since the authors mention iron-sulfur clusters in complex II, does complex II relate to ferroptosis?
2. Do mutations in complex II cause distinct phenotypes of Leigh's syndrome comparing to mutations in other complexes?
3. The authors should select better references to make the claim that TNF and cytokines inhibit complex II (Ref.43). The original reference show the activities of most complexes all decreased, but not specifically for complex II.
4. It may not be appropriate to cite Ref. 294 to suggest that malonate promotes proliferation of adult cardiomyocytes, since human cardiomyocytes are considered to be not able to proliferate.
5. Paragraph 4 is interesting, however the authors should provide more link between complex II with this paragraph. Is manipulation of activities or protein level of complex II contribute to those examples?
Author Response
- Since the authors mention iron-sulfur clusters in complex II, does complex II relate to ferroptosis?
Response: That is a very interesting question that we had not considered. Upon searching the literature, it appears that complex II is not known to contribute to ferroptosis. A recent article (https://biosignaling.biomedcentral.com/articles/10.1186/s12964-022-01025-9) nicely reviews the literature in this area and implicates Complex III, not II, as a contributor to ferroptosis.
- Do mutations in complex II cause distinct phenotypes of Leigh's syndrome comparing to mutations in other complexes?
Response: This is another interesting question. It appears that Complex II does not cause a distinct phenotype relative to other forms of Leigh’s syndrome, at least not that can be discerned from the small number of patients identified thus far. Leigh’s syndrome is a highly heterogeneous collection of symptoms, with variable progression, and complex II deficiency is a rare cause of Leigh’s compared to other genes as reviewed here in 2020: https://ojrd.biomedcentral.com/articles/10.1186/s13023-020-1297-9
- The authors should select better references to make the claim that TNF and cytokines inhibit complex II (Ref.43). The original reference show the activities of most complexes all decreased, but not specifically for complex II.
Response: Yes, we agree that this section should be more clear that cytokines not only suppress Complex II, but other complexes as well. We have rewritten this paragraph (highlighted text, starts at line 225) and added additional references.
- It may not be appropriate to cite Ref. 294 to suggest that malonate promotes proliferation of adult cardiomyocytes, since human cardiomyocytes are considered to be not able to proliferate.
Response: Reference 69 demonstrates the remarkable finding that dimethyl malonate can induce adult cardiomyocytes to re-enter the cell cycle, in vivo. We have clarified and expanded on this observation in the highlighted text beginning on line 296.
- Paragraph 4 is interesting, however the authors should provide more link between complex II with this paragraph. Is manipulation of activities or protein level of complex II contribute to those examples?
Response: The answer to this question is certainly yes, in that any factor that modulates complex II will alter succinate levels. We have improved the transition from section 3 into section 4 by adding sentences to emphasize that the effects of succinate described in this section can all be downstream of altered complex II activity (text is highlighted).
Reviewer 2 Report
Mitochondrial medicine is a very important part of biomedical research. For this reason, the work is significant. The work is clearly written and easy to read. The additional two figures make it easier for the reader to understand the text. Similarly, the division of the text into chapters according to diseases increases comprehensibility. In the work, the authors cited works from recent years. I have no fundamental comments on the text. Just a small reminder - check the text to see if all used abbreviations are explained in the text the first time they are used.Author Response
Response to Reviewer #2
Thank you for your kind words about our manuscript.
In response to your comment, we have double-checked that abbreviations are explained upon first usage.
Round 2
Reviewer 2 Report
The authors explained the abbreviations used, Thank you.